# Sharpness Can Be Manipulated and Misleading for Generalization

## Abstract

Sharpness, commonly measured by the Hessian matrix, has long been hypothesized to correlate with generalization. However, this work presents several counterexamples where Hessian-based sharpness can be manipulated. We derive a formula for the Hessian trace, revealing its dependence on several key factors: the norm of network weights, activation frequency, and the entropy of the output distribution. By manipulating these factors, we construct scenarios where models reside in flat minima yet exhibit overfitting and poor test performance. Moreover, Gaussian noise injection reduces Hessian trace within the first epoch and can even yield arbitrarily flat minima without corresponding improvement in generalization. This suggests that sharpness may be correlated with generalization without being causally responsible for it.

## 1 Introduction

The remarkable success of neural networks stems from their ability to learn meaningful representations and generalize to unseen data, rather than merely memorizing training examples. Poor generalization, commonly known as overfitting is a key issue to be avoided. The simplest metric for assessing generalization is the gap between training and test accuracy.

Over time, research has shifted toward analyzing the intrinsic geometry of the loss landscape itself instead of focusing on differences in accuracy, aiming to understand generalization through the optimization dynamics of gradient descent. For a long time, it has been widely believed that flat minima correlate with better generalization Jastrzębski et al. (2017); Taki (2017); Murata et al. (1994).

However, several counterexamples challenge this view. Some studies have shown that guiding optimization into sharp minima via anisotropic learning rates does not necessarily lead to poor generalization Jastrzębski et al. (2019). Others have demonstrated that scaling the network weights $W$ by a constant factor $\alpha$ can arbitrarily alter the sharpness without affecting generalization Dinh et al. (2017). We reproduce this result in Figure 1. Moreover, some works have even observed a negative correlation between sharpness and generalization in certain settings Andriushchenko et al. (2023).

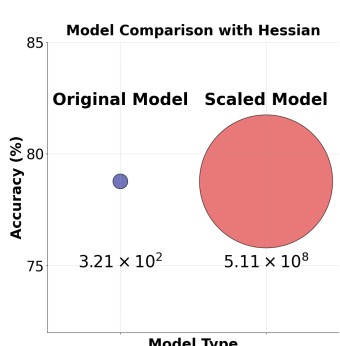

Figure 1: ResNet-18 on CIFAR-10 for 10 epochs. After training all weights are scaled $\times 10$. Train Acc = 90.29% Test Acc = 78.77% and remain unchanged after scaling. However, the largest Hessian eigenvalue increases from $3.21 \times 10^2$ to $5.11 \times 10^8$. The vertical axis shows test accuracy and the size of each circle represents the magnitude of the largest Hessian eigenvalue.

Therefore, we are led to a critical question: *Is Hessian-based sharpness truly a reliable indicator of generalization?*

Our study is organized into several key components. First, we derive two analytical expressions for the trace of the Hessian matrix: the *Scale-Aware Upper Bound* and the *Centered Decomposition*. These formulas provide theoretical guidance for constructing counterexamples to the commonly as-

sumed relationship between sharpness and generalization. Through experiments, we demonstrate that sharpness is implicitly influenced by several factors: the scale effects induced by activation frequency variations and the probability distribution of the softmax outputs in the final layer (entropy). By manipulating these factors, we are able to construct models that exhibit flat minima yet generalize poorly.

Further, we observe that sharpness decreases in the first epoch with noise injection. Moreover, we can also construct scenarios in which sharpness decreases under noise injection, yet generalization performance remains unchanged. This provides new insights into the discussion of whether the relationship between sharpness and generalization is causal or correlational.

## 2 BACKGROUND

**Sharpness of the Loss Landscape:** We monitor the sharpness of the loss landscape using two widely adopted metrics: the largest eigenvalue of the Hessian matrix (stochastic power iteration) and the Hessian trace (Hutchinson random probing). Both measures have been extensively used in prior work Foret et al. (2021); Jastrzębski et al. (2019); Sagun et al. (2018). We leverage PyTorch's automatic differentiation capabilities to compute Hessian-vector products on the fly to avoid explicitly constructing or storing the full Hessian matrix. All monitoring is on the training set consistent with other research.

**Generalization Performance:** Generalization is assessed by comparing test accuracy under matched training accuracy conditions, with models trained until training accuracy reaches at least 90%, guaranteeing that optimization has converged sufficiently close to a local minimum.

**Activation Frequency and Death Frequency:** Here, activation frequency is mentioned because it naturally induces changes in weight norms through training dynamics, unlike artificial scaling which is manually imposed. We define the *activation frequency* as follows. Given $B$ samples, for each individual sample, the activation at layer $l$ is $\mathbf{a}^{(l)} = \text{ReLU}(\mathbf{z}^{(l)})$, $\mathbf{z}^{(l)} = \mathbf{W}^{(l)}\mathbf{a}^{(l-1)} + \mathbf{b}^{(l)}$. $N$ denotes the number of elements in $\mathbf{a}^{(l)}$ and $M_i$ is the number of entries in $\mathbf{z}^{(l)}$ which become zero after applying ReLU. The activation frequency is: $\eta = \frac{1}{B}\sum_{i=1}^{B}\left(1 - \frac{M_i}{N}\right) = 1 - \frac{1}{B}\sum_{i=1}^{B}\frac{M_i}{N}$. *Death frequency* is the proportion of output dimensions that are *consistently* zeroed across the entire dataset of $B$ samples. Let $P$ denote the number of output positions that remain zero for all $B$ inputs at a given layer, the death frequency is: $\xi = \frac{P}{N}$.

Note that this notion of "death" refers to permanently inactive output dimensions, fundamentally different from pruning. In our analysis, activation frequency is computed *excluding* such dead outputs.

**Experimental Setup:** We focus on image classification networks, as the initial studies on the relationship between generalization and sharpness were conducted in this setting. In our experiments, we introduce a custom VGG-small architecture (see Appendix B.2) that excludes batch normalization and residual connections, preserving a basic chain of matrix multiplications to minimize interference from confounding architectural factors. To improve the generalizability of our findings, we also complement the results with additional experiments across diverse datasets and model architectures.

For all VGG-small experiments, we use the SGD optimizer with a learning rate of 0.01 and momentum 0.9. All experimental variants are based on SGD as well, differing only in the learning rate. We intentionally avoid adaptive optimizers such as Adam, which implicitly incorporate curvature-like information through their adaptive learning rates. Since the interaction between such methods and sharpness is not yet fully understood, their use could confound our analysis.

All metrics are evaluated on the **clean, unmodified dataset** to ensure consistent comparisons.

# 3 ON THE STRUCTURE OF THE GAUSS-NEWTON HESSIAN IN NEURAL NETWORKS

The full derivation is provided in Appendix A. While the expression applies to a single sample, in practice the Hessian trace is estimated as an empirical average across multiple batches. We omit the expectation operator for notational simplicity.

## 3.1 NOTATION

1. $f(\theta; x) \in \mathbb{R}^C$: the network output for input $x$ under parameters $\theta$ before softmax, $p = softmax(f(\theta; x))$.

2. $a^{(l-1)}$: the post-activation at layer $l$, $s_{l-1} = \|a^{(l-1)}\|_2^2$.

3. $\delta_i^{(L)} = \frac{\partial f_i}{\partial z^{(L)}} \in \mathbb{R}^{n_L}$: the gradient of the $i$-th output logit $f_i$ with respect to the pre-activation vector $z^{(L)}$ at the output layer. This vector captures how the value of $f_i$ depends on each component of the input to the softmax, and is used in backpropagation to propagate gradients into the network, and $v_i = \delta_i^{(L)}$.

4. $M^{(l)} \in \mathbb{R}^{n_L \times n_L}$: a positive semi-definite matrix that captures the backpropagation dynamics from layer $l$ to the output layer, defined as $M^{(l)} = \left( \overleftarrow{\prod_{k=l}^{L-1}} W^{(k+1)} \tilde{D}^{(k)} \right) \left( \overleftarrow{\prod_{k=l}^{L-1}} \tilde{D}^{(k)} W^{(k+1)\top} \right)$, where $\tilde{D}^{(k)} = \mathrm{Diag}(\sigma'(z^{(k)}))$ is the diagonal matrix of activation derivatives at layer $k$, and the left arrow denotes reverse-order matrix multiplication.

## 3.2 SCALE-AWARE UPPER BOUND

Applying Jensen's inequality and bounding via the spectral norm $\lambda_{\max}(M^{(l)})$, we obtain the following upper bound:

$$\mathrm{Tr}(H_\theta) \leq 2 \sum_{l=1}^{L} \|a^{(l-1)}\|_2^2 \cdot \lambda_{\max}\left(M^{(l)}\right) \cdot \sum_{i=1}^{C} p_i \|\delta_i^{(L)}\|^2$$

Although this upper bound is relatively loose, it remains informative. It consists of four key components: (1) the squared $\ell_2$-norm of forward activations $\|a^{(l-1)}\|_2^2$ (2) the squared $\ell_2$-norm of output-layer gradient signals $\|\delta_i^{(L)}\|^2$ (3) the largest eigenvalue of the path matrix $\lambda_{\max}(M^{(l)})$ (4) the anisotropic scaling weights $p_i$, derived from the softmax probability distribution.

If the Frobenius norm of the weight matrices $W^{(k)}$ decreases and network sparsity increases (more ReLU units become inactive), then the activation norms $\|a^{(l-1)}\|_2^2$ will generally decrease, thereby lowering the entire upper bound. Moreover, for the matrix $M^{(l)}$, sparsity directly affects the mask matrices $\tilde{D}^{(k)}$, while weight scaling affects $W^{(k+1)}$, both may indirectly influence $\lambda_{\max}(M^{(l)})$. This illustrates the role of scale in shaping the Hessian spectrum, which can also explain why scaling the weights leads to changes in the sharpness (Figure 1), as it alters the upper bound.

## 3.3 CENTERED DECOMPOSITION

$$\mathrm{Tr}(H_\theta) \approx \sum_{l=1}^{L} s_l \left( \sum_{i=1}^{C} p_i \cdot v_i^\top M^{(l)} v_i - \sum_{i,j=1}^{C} p_i p_j \cdot v_i^\top M^{(l)} v_j \right)$$

The Hessian is the derivative of the gradient. In neural networks, the backpropagated gradient involves each weight matrix $W$ in its transposed form $W^\top$, creating a tight coupling between forward and backward dynamics.

Moreover, let $\bar{v} = \sum_i p_i v_i$. We begin with a simple rearrangement:

$$\mathrm{Tr}(H_\theta) \approx \sum_{l=1}^{L} s_{l-1} \left( \sum_i p_i v_i^\top M^{(l)} v_i - \bar{v}^\top M^{(l)} \bar{v} \right) = \sum_{l=1}^{L} s_{l-1} \sum_i p_i (v_i - \bar{v})^\top M^{(l)} (v_i - \bar{v})$$

This form is particularly insightful: it resembles a *Mahalanobis-like distance*. If $M^{(l)}$ were the identity, this would reduce to the weighted variance of gradients across classes. Here, $s_{l-1} = \|a^{(l-1)}\|_2^2$ measures the *strength of contribution* from layer $l$, $p_i$ reflects the *importance* (probability) of class $i$, $(v_i - \bar{v})^\top M^{(l)}(v_i - \bar{v})$ quantifies how strongly class $i$ activates the curvature pathway at layer $l$.

In the following section, we will conduct studies based on both the upper bound and this centered decomposition.

## 4    NUMERICAL IMPACT OF ACTIVATION SPARSITY ON THE HESSIAN

The derived Scale-Aware Upper Bound reveals that, unlike prior studies which focus solely on parameter scale, the norms of all components including weights, activations, inputs, and gradients play a crucial role in shaping the loss landscape. Specifically, the sparsity introduced by ReLU leads to zeroed-out forward activations, reduces the overall norm, thereby lowering this upper bound and ultimately results in decreased sharpness. We will now demonstrate this effect.

### 4.1    THE IMPACT OF ACTIVATION FREQUENCY ON HESSIAN

We first show that changes in activation frequency indeed affect the eigenvalues of the Hessian. Here, we present only a numerical correlation as preliminary evidence to motivate the subsequent experiments.

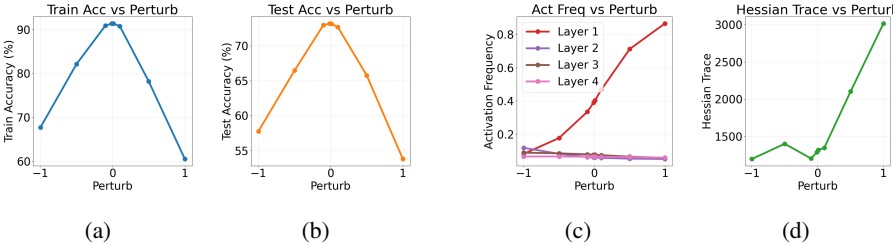

Figure 2: Performance after 7 epochs of training on VGG-small with additive perturbations applied to the first layer's bias. From left to right: accuracy on training a and test b dataset, c activation frequency of each layer, d Hessian trace of the network.

Here, we manipulate the activation frequency by adding perturbations to the original bias of the first layer, as shown in Figure 2. When the perturbation increases from -1 to 1, the activation frequency rises, and correspondingly, the Hessian trace also increases.

This pattern is artificially induced. Nevertheless, similar patterns of coordination may arise spontaneously during training, although the underlying mechanism remains unclear (see Appendix A.2).

### 4.2    OVERFITTING WITH SMALL HESSIAN EIGENVALUES VIA FEATURE MAP MASKING

We construct a scenario in which smaller Hessian eigenvalues coincide with stronger overfitting, aiming to investigate the role of dead outputs in this phenomenon. The relationship between activation frequency and generalization has not been systematically studied; it has only been briefly mentioned or assumed in passing in prior work Evci et al. (2022); Hooker et al. (2021).

We therefore shift our focus to output death. Due to the non-differentiability of ReLU in the negative half-space, activations can become permanently zero Lu et al. (2020). As more output units enter this inactive state, the model's expressive capacity becomes increasingly constrained, limiting its ability to capture diverse patterns in the data.

One potential cause of output death is improper initialization or inappropriate learning rates Glorot & Bengio (2010). However, in convolutional layers, it is difficult to induce significant ReLU death without modifying the architecture, and such methods inevitably affect other aspects of training. Moreover, the natural rate of ReLU death in standard convolutional layers is relatively low. To

reliably study its effects, we directly apply a fixed masking ratio to the feature maps and keep it unchanged throughout training.

As shown in Table 1, the masked model exhibits clear overfitting, yet has a smaller maximum Hessian eigenvalue than the unmasked counterpart. Additional results on ResNet-18 are provided in Appendix A.3.

Table 1: Performance comparison between Mask (70%) and UnMask versions on VGG-small.

| Version | Train Acc (%) | Test Acc (%) | $H\lambda_{\max}$ |
|---------|---------------|--------------|-------------------|
| Mask 70% | 89.17 | 64.37 | **151.6612** |
|          | 90.58 | 65.72 | **127.9235** |
| UnMask   | 91.43 | 78.68 | **317.2083** ↑ |
|          | 88.43 | 79.11 | **350.7079** ↑ |

## 5 IMPACT OF THE SOFTMAX DISTRIBUTION ON THE HESSIAN

In our previous derivation, we have $\text{Tr}(H_\theta) \propto \sum_i p_i (v_i - \bar{v})^\top M^{(l)} (v_i - \bar{v})$, the scaling factor $s$ is omitted. To analyze the structural behavior, we consider the simplified case where $M^{(l)}$ is set to the identity matrix $I$. Under this assumption, we have:

$$\text{Tr}(H_\theta) \propto \sum_i p_i \|v_i - \bar{v}\|^2 = 1 - \|p\|_2^2$$

which serves as a proxy for distributional uncertainty, closely related to Gini impurity or the complement of Shannon entropy's quadratic approximation (detailed derivation is in Appendix A.1).

This leads to a remarkably concise conclusion: the variance across class gradients is directly determined by the dispersion of the predicted probability distribution, higher uncertainty in predictions corresponds to higher variance (the term "variance" used here as a formal construct, not a statistical estimate).

To further characterize this relationship, we examine its behavior at extremal points:

1. Maximum variance occurs when the predicted distribution $p_i = 1/C$.

2. Minimum variance occurs when the predicted distribution is deterministic (one-hot) $p_k = 1$ for some class $k$ and $p_j = 0$ for $j \neq k$.

This implies that the sharpness is intrinsically linked to the entropy or certainty of the model's output distribution. As the softmax output approaches a one-hot vector, the landscape becomes flatter. In fact, the simplest way to construct region-wise one-hot outputs is to apply a temperature factor to the output softmax. This approach however overlaps with our previous norm-scaling experiments. We instead explore two alternative methods to create this scenario.

### 5.1 DISENTANGLING CONFOUNDING FACTORS IN FLATNESS-GENERALIZATION OBSERVATIONS

We begin by describing the experimental setup. Since the importance of network norms has been previously discussed, the following experiments strictly control these norms. We use a baseline to illustrate how network norms differ during training when data augmentation is applied, as shown in Figure 3. Evidently, when the training accuracy reaches around $90\%$, the data-augmented group has a higher test accuracy, indicating better generalization, and the corresponding maximum Hessian eigenvalue is also smaller. We can say that the network has fallen into a flat minimum at this point.

However, we observed that in the data augmentation group, both the convolutional parameter norms and the forward activation norms were larger, while activation frequencies were smaller. Concurrently, the Hessian becomes **incomparable** in this scenario. Therefore, we impose norm constraints on the entire neural network by controlling the mean $\mu$ and variance $\sigma^2$. During each forward pass, the activations of each layer are normalized to have $\mu = 0$ and $\sigma^2 = 1$. For the weight matrix $W$, we constrain it to have $\mu = 0$ and $\sigma^2 = \frac{2}{n_{\text{in}}}$, consistent with He initialization He et al. (2015a).

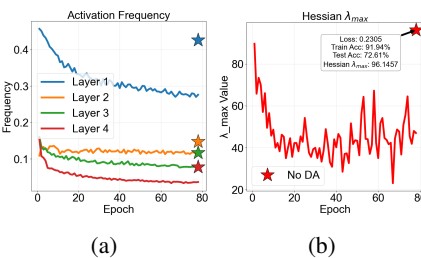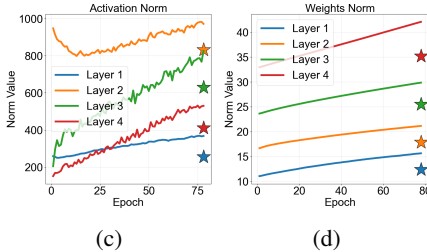

|     (a)     |     (b)     |     (c)     |     (d)     |

Figure 3: Training dynamics over 80 epochs for the data augmentation group (`RandomCrop` and `RandomHorizontalFlip`). From left to right: a activation frequency, b maximum Hessian eigenvalue, c activation norm, and d weight norm. The stars indicate measurements from the no-data-augmentation group at epoch 11, where training accuracy reached 91.84% and test accuracy was 72.00%. For comparison, the data augmentation group achieves 90.56% training accuracy and 83.07% test accuracy at epoch 80.

In contrast to batch-based normalization methods, our method is applied independently for each sample.

This approach brings two key benefits. First, variance constraints implicitly control the norm. For the weight matrix $W$ we have: $\mathbb{E}[w_i^2] = \frac{2}{n_{\text{in}}}$. The expected squared Frobenius norm of $W$ is: $\mathbb{E}[\|W\|_F^2] = \sum \mathbb{E}[w_i^2] = n_{\text{in}} \cdot n_{\text{out}} \cdot \frac{2}{n_{\text{in}}} = 2n_{\text{out}}$. A similar argument applies to the forward activations. Second, zero mean ensures activation sparsity around 50%.

## 5.2 Cycling Through Training Batches can Affect Sharpness

Table 2: Hessian monitoring of overfitting experiments on VGG-small. Where $N$ is the proportion of shuffled labels, and $R$ is the number of repeated training of each batch. "DA" means Data Augumentation. The first three rows in the table show the results after the first training epoch.

| Settings | Train Acc (%) | Test Acc (%) | Tr(H) | $\lambda_{\max}(\mathbf{H})$ | $1 - \|p\|_2^2$ |
|---|---|---|---|---|---|
| $R = 5, N = 0$ | 54.10 | 52.14 | **500.1708** | **29.0367** | 0.774225 |
| $R = 10, N = 0$ | 55.01 | 45.21 | **1017.1961** | **60.6709** | 0.766617 |
| $R = 20, N = 0$ | 45.89 | 44.79 | **2370.0525** | **137.2921** | 0.775230 |
| $R = 1, N = 0$ (no DA) | 88.55 | 67.98 | **2026.9998** | **63.0798** | 0.659916 |
| $R = 1, N = 0$ (DA) | 89.68 | 83.42 | **719.5708** | **38.0442** | 0.555853 |
| $R = 5, N = 0$ | 89.89 | 77.03 | **695.0275** | **19.6098** | 0.604527 |
| $R = 10, N = 0$ | 91.83 | 77.03 | **570.7282** | **22.3277** | 0.562706 |

We conduct the following experiment: during training, we cyclically train on a single batch for $R$ iterations instead of only one pass. According to our theory, it is reasonable to hypothesize that in the early stages of training, the model tends to memorize the data, leading to a strong dependence between inputs and outputs and consequently a sharper loss landscape. However, as training progresses, correct learning can still occur as provided $R$ is not excessively large. Repeated cycling on the same batch pushes the model's output distribution toward a one-hot vector even if generalization performance remains unchanged, results are shown in Table 2.

Clearly, at the first epoch, although R increases from 5 to 20 and sharpness also rises, $1 - \|p\|_2^2$ remains unchanged. When training progresses to around 90% of the total epochs, increasing $R$ leads to lower sharpness and $1 - \|p\|_2^2$ decreases accordingly. However, the reason why the data augmentation group exhibits even lower values remains, possibly suggesting more complex interactions at play.

More detailed training dynamics on ResNet-18 and CIFAR-10 are shown in Figure 4.

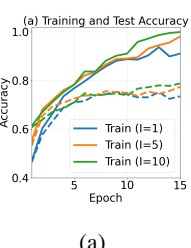 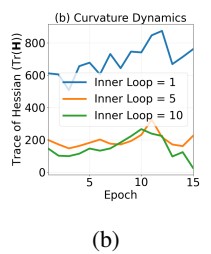 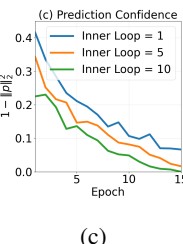

(a)                 (b)                 (c)

Figure 4: Experiment on ResNet-18 trained on CIFAR-10 using SGD (lr $= 0.01$, momentum $= 0.9$). The original BatchNorm layers are replaced with strict per-sample mean-variance normalization. Results are shown for the first 15 epochs. From left to right: a training and test accuracy, b Trace of Hessian, c $1 - \|p\|_2^2$.

### 5.3 The Impact of Data Leakage on Sharpness

To further investigate, we construct an experiment with data leakage. Data leakage causes the network to rely only on certain parts of the input, leading to overconfident outputs that tend toward one-hot vectors. This represents an extreme case of overfitting. Importantly, such scenarios are not merely artificial: dataset bias often acts as a form of implicit data leakage in real-world settings. For instance, a network might classify based on zebra stripes rather than global features Selvaraju et al. (2019), or classify bananas by color instead of shape Niu et al. (2021). Since it is difficult to construct controlled experiments with biased datasets, we instead study the data leakage setting here.

To this end, we design the following experiment:

**1. Experimental group**: For each of the 10 classes, a fixed MNIST image is selected, resized to 8×8 and pasted onto the bottom-left corner of every corresponding CIFAR-10 training image. This introduces a deterministic, class-specific data leakage.

**2. Control group**: An 8×8 MNIST patch is similarly pasted in the same location, but resampled randomly for each batch. This preserves pixel-level variation while preventing consistent associations between the patch and class labels.

This design ensures that any observed differences in the Hessian arise from the *leakage of label information*, rather than from the presence of static pixel patterns. The results are shown in Table 3. Norm constraints are still applied throughout training.

Table 3: Data leakage experiment on VGG-small: monitoring at the epoch of first reaching 100% training accuracy. Rows correspond to the data leakage group and the control group; columns report three sensitivity metrics: Trace of the Hessian ($\mathrm{Tr}(\mathbf{H})$), Frobenius norm $\|\mathbf{J}_x\|_F$ of the input-output Jacobian over all pixels, $\|\mathbf{J}_x\|_F$ with an $8 \times 8$ additive patch region excluded, and $1 - \|p\|_2^2$. The excluded value is rescaled to match the spatial support of an $8 \times 8$ region using: $\sqrt{\frac{\|\mathbf{J}_x(\text{excluded})\|_F^2}{32 \times 32 - 8 \times 8}} \times \sqrt{8 \times 8}$ which enables a fair comparison between the effective sensitivity in the remaining pixels and that expected from a small, localized pattern.

| Settings | $\mathrm{Tr}(\mathbf{H})$ | $\|\mathbf{J}_x\|_F$ (all pixels) | $\|\mathbf{J}_x\|_F$ (excluded) | $1 - \|p\|_2^2$ |
|---|---|---|---|---|
| Data Leakage Group | 15.22927 | 69.61466 | 20.98831 | 0.00046 |
| Control Group | 52.55664 | 22.88224 | 23.17986 | 0.00154 |

Notice that for the data leakage group, the $1 - \|p\|_2^2$ metric is lower, and correspondingly the sharpness is also lower. Here, we further introduce the Frobenius norm of the input-output Jacobian matrix, which have been widely associated with generalization in many studies Novak et al. (2018); Sokolić et al. (2017); Johansson et al. (2022). For the data leakage group, there is a noticeable difference between excluding and not excluding the $8 \times 8$ region whereas the control group shows little change, suggesting the network exhibits selectivity over specific input regions. Moreover, the Jacobian norm across all input pixels is larger in the data leakage group than in the control group, in-

dicating higher sensitivity across the full input space. In such cases, the Jacobian norm (input-output sensitivity) may be a more meaningful indicator than sharpness.

# 6 Flatness from Noise Injection: Correlation or Causation?

## 6.1 Noise Injection Reduces Sharpness in Early Training

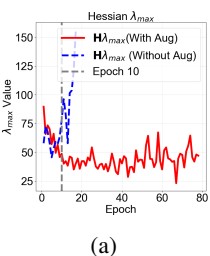

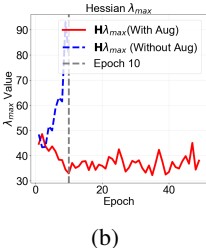

(a)                                                                 (b)

Figure 5: Training dynamic on VGG-small. a and b: Maximum eigenvalue of the Hessian without and with normalization. The dashed vertical lines indicate the epoch at which the non-data-augmented model reached approximately 90% training accuracy.

Recalling our previous normalization operation, here we plot the evolution of the Hessian Trace during training, with results shown in Figure 5. An interesting phenomenon emerges. The gray vertical line indicates the epoch at which the training accuracy reaches 90%. For the standard group, the Hessian traces under normal training and data augmentation remain closely aligned before this point, only diverging significantly afterward. In contrast, when the modified normalization is applied the divergence occurs earlier. This observation prompts further reflection.

Many studies have focused on how data augmentation affects sharpness, concluding that it suppresses the growth of sharpness and smooths the loss landscape LeJeune et al. (2019); Yoo & Yoon (2025); Zhang et al. (2024). However, these conclusions typically rest on the assumption that flatness causes better generalization. Yet, even if causality cannot be entirely dismissed, could the observed relationship instead be primarily driven by correlation? To explore this, we consider a simpler and more quantifiable data augmentation scenario: injecting Gaussian noise. This strategy has been adopted in multiple works and is regarded as an effective form of data augmentation Zhang et al. (2024); Yuan et al. (2023); Lopes et al. (2019). Adding Gaussian noise effectively places each data point at the center of an isotropic Gaussian ellipsoid with radius determined by the noise scale. Through the network's nonlinear transformation, this ellipsoid maps into the loss landscape and lowers loss across its neighborhood (detailed explanation can be found in Appendix A.5). Given this perspective, it becomes plausible that in such cases, the connection between flatness and generalization may be influenced more by correlation than direct causation.

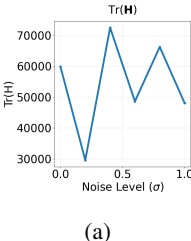

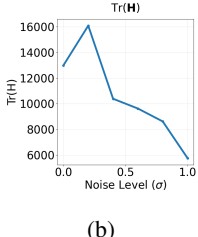

(a)                                                                 (b)

Figure 6: VIT-Base experiment on a subset of ImageNet, trained with SGD (lr = 1e-4, momentum = 0.9). We monitor the trace of the Hessian at the end of the first epoch as the variance of injected Gaussian noise increases. a shows the standard model, while b uses per-sample mean-variance normalization applied strictly across all layers.

In other words, could data augmentation (here limited to noise injection) reduce sharpness as early as the first epoch, rather than leading to flat minima only in later stages? To investigate this, we injected

Gaussian noise with varying variances into the inputs and measured the Hessian trace, as shown in Figure 6a. Notably, under standard settings, the Hessian trace exhibited irregular oscillations without a clear trend. However, as previously discussed, network norm significantly influences sharpness estimation. We therefore replaced the original LayerNorm in ViT with strict variance-mean normalization, as illustrated in Figure 6b. Under this modified setup, a clear trend appears: as the injected noise increases, Tr(**H**) gradually decreases. In fact, when the noise variance exceeds 0.3, training becomes impossible yet sharpness continues to decrease. Additional experimental results are provided in Appendix A.4.

## 6.2 SHARPNESS REDUCTION VIA NOISE INJECTION CAN BE UNRELATED TO GENERALIZATION

Table 4: Hessian trace monitoring experiment for ResNet-18 (without mean-variance normalization) on CIFAR-100 using SGD with learning rate 0.01 and momentum 0.9. All evaluations are performed on the clean training set.

| Group | Train Acc (%) | Test Acc (%) | Tr(H) |
|---|---|---|---|
| Standard | 89.22 | 72.37 | **7391.83** ↑ |
| Noise Finetune | 90.19 | 71.66 | **4650.93** |

Based on our observations and analysis of noise injection, we construct another counterexample to decouple generalization from sharpness. We first train a ResNet-18 on CIFAR-100 to achieve 99% training accuracy. Then, in one additional final epoch, we inject input noise with variance 0.5 (such noise hinders training from scratch, preventing it from acting as effective data augmentation and thus helping to maintain a consistent generalization gap). Results are shown in Table 4.

After this noisy epoch, the training accuracy drops to 90%, yet the test accuracy remains nearly unchanged compared to the standard training counterpart. However the two models exhibit significantly different Hessian traces. This demonstrates that under noise injection, sharpness can be reduced without any corresponding change in generalization, further suggesting that sharpness **can** not be a causal driver but rather a correlative artifact.

## 7 CONCLUSION

The idea that flat minima correspond to good generalization is both intuitively appealing and often associated with principles in PAC-Bayes theory. We do not claim this perspective is necessarily wrong. However, in this work, we demonstrate through theoretical derivation and empirical evidence that, within the deep learning framework, sharpness, at least Hessian-based sharpness, can be manipulated. The overall norm of the network and the output distribution after the final softmax layer both influence the Hessian trace. Injecting noise can induce spurious flatness, giving the appearance of a flat minimum without actual improvement in generalization. This raises the possibility that, in scenarios involving data augmentation, the observed flatness may result from implicit noise injection. Consequently, the correlation between flatness and generalization performance under data augmentation may reflect association rather than causation.

## 8 LLM DECLARATION

Large language models were used for writing refinement and literature review assistance. All other contributions including conceptualization, methodology design, experimentation, data analysis, and interpretation, were carried out by the human authors.

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

# A  APPENDIX

## A.1  DETAILED DERIVATION OF THE UPPER BOUND FOR THE TRACE OF THE HESSIAN

Consider a single layer $l$ in a deep network, with pre-activation $z^{(l)} = W^{(l)}a^{(l-1)}$ (To simplify, we won't consider the bias term here) and activation $a^{(l)} = \text{ReLU}(z^{(l)})$. $\overleftarrow{\prod_{k=l}^{L-1}} \left( \tilde{D}^{(k)}W^{(k+1)\top} \right) :=$ $\tilde{D}^{(L-1)}W^{(f)\top}\tilde{D}^{(L-2)}W^{(L-1)\top} \cdots \tilde{D}^{(l)}W^{(l+1)\top}$. The output of $f$ with respect to $z^{(l)}$, denoted $\delta^{(l)} = \partial f/\partial z^{(l)}$, is computed via backpropagation using the chain rule: $\delta^{(l)} = \frac{\partial f}{\partial z^{(l)}} = \frac{\partial f}{\partial z^{(l+1)}} \cdot \frac{\partial z^{(l+1)}}{\partial a^{(l)}} \cdot \frac{\partial a^{(l)}}{\partial z^{(l)}}$

$$\delta^{(l)} = \left( W^{(l+1)\top}\delta^{(l+1)} \right) \odot \sigma'(z^{(l)}) = \left( \overleftarrow{\prod_{k=l}^{L-1}} \text{Diag}(\sigma'(z^{(k)}))W^{(k+1)\top} \right) \delta^{(L)}$$

The output of $f$ with respect to $W^{(l)}$ is: $\nabla_{W^{(l)}} f = \frac{\partial f}{\partial W^{(l)}} = \frac{\partial f}{\partial z^{(l)}} \cdot \frac{\partial z^{(l)}}{\partial W^{(l)}} = \delta^{(l)} \cdot a^{(l-1)\top}$

Considering a single sample and the parameters of one layer, $\nabla_{W^{(l)}} f = \delta^{(l)} a^{(l-1)\top}$, we have:

$$\|\nabla_{W^{(l)}} f\|_F^2 = \|\delta^{(l)}\|_2^2 \cdot \|a^{(l-1)}\|_2^2$$

Substitute the path expression for $\delta^{(l)}$, where $\tilde{D}^{(k)} = \text{Diag}(\sigma'(z^{(k)}))$ serves as a mask matrix, using the sub-multiplicativity property, we have:

$$\|\delta^{(l)}\|_2^2 \leq \left\| \overleftarrow{\prod_{k=l}^{L-1}} \tilde{D}^{(k)} \cdot W^{(k+1)\top} \right\|_2^2 \cdot \|\delta^{(L)}\|_2^2$$

For the Hessian matrix, which is a matrix of second-order derivatives, direct computation is often intractable. Fortunately, a well-known Gauss-Newton decomposition exists:

$$\nabla_\theta^2 \mathcal{L} = \underbrace{J_\theta^\top \cdot \nabla_y^2 \mathcal{L} \cdot J_\theta}_{\text{Gauss-Newton term}} + \underbrace{\sum_{i=1}^C \frac{\partial \mathcal{L}}{\partial y_i} \cdot \nabla_\theta^2 f_i(\theta)}_{\text{residual term}}$$

where the first term is a quadratic form involving the Jacobian matrix, and the second term is the residual term. Numerous studies Lee et al. (2023); Fort & Jastrzebski (2019) have shown that in neural networks the Gauss-Newton term dominates, especially those with ReLU activations. Therefore, in the following derivation, we approximate the Hessian by retaining only the first term $\text{Tr}(\nabla_\theta^2 \mathcal{L}) \approx \text{Tr}(J_\theta^\top H_L J_\theta)$, where $H_L = \nabla_y^2 \mathcal{L}$. By the cyclic property of the trace, we have: $\text{Tr}(J_\theta^\top H_L J_\theta) = \text{Tr}(H_L J_\theta J_\theta^\top)$. Define the Gram matrix $G = J_\theta J_\theta^\top \in \mathbb{R}^{m \times m}$, $\text{Tr}(\nabla_\theta^2 \mathcal{L}) \approx \text{Tr}(H_L \cdot G)$.

When using softmax with cross-entropy loss, $H_L = \nabla_y^2 \mathcal{L} = \text{diag}(p) - pp^\top$, where $p \in \mathbb{R}^C$ is the predicted probability vector. Substituting this into the trace expression gives:

$$\text{Tr}(\nabla_\theta^2 \mathcal{L}) \approx \text{Tr}\left( (\text{diag}(p) - pp^\top) \cdot G \right) = \sum_{i=1}^C p_i G_{ii} - \sum_{i=1}^C \sum_{j=1}^C p_i p_j G_{ij} = \sum_{i=1}^C p_i G_{ii} - p^\top G p$$

Since $G_{ii} = \|\nabla_\theta f_i\|^2$, we arrive at the final expression:

$$\boxed{\text{Tr}(H_\theta) \approx \sum_{i=1}^C p_i \cdot \|\nabla_\theta f_i\|^2 - p^\top G p}$$

For any pair of output dimensions $i, j$, the inner product of their parameter gradients is:

$$[JJ^\top]_{ij} = \langle \nabla_\theta f_i, \nabla_\theta f_j \rangle = \sum_{l=1}^L \langle \nabla_{W^{(l)}} f_i, \nabla_{W^{(l)}} f_j \rangle$$

For fully connected layers, $\nabla_{W^{(l)}} f_i = \delta_i^{(l)} a^{(l-1)\top}$, hence:

$$\langle \nabla_{W^{(l)}} f_i, \nabla_{W^{(l)}} f_j \rangle = \mathrm{Tr}\left((\delta_i^{(l)} a^{(l-1)\top})^\top (\delta_j^{(l)} a^{(l-1)\top})\right) = (a^{(l-1)\top} a^{(l-1)}) \cdot (\delta_i^{(l)\top} \delta_j^{(l)})$$

$$\boxed{[JJ^\top]_{ij} = \sum_{l=1}^{L} \|a^{(l-1)}\|_2^2 \cdot \langle \delta_i^{(l)}, \delta_j^{(l)} \rangle}$$

$$\langle \delta_i^{(l)}, \delta_j^{(l)} \rangle = \delta_i^{(L)\top} M^{(l)} \delta_j^{(L)} \quad M^{(l)} = \left(\overleftarrow{\prod_{k=l}^{L-1}} W^{(k+1)} \tilde{D}^{(k)}\right) \left(\overrightarrow{\prod_{k=l}^{L-1}} \tilde{D}^{(k)} W^{(k+1)\top}\right)$$

$$\mathrm{Tr}(H_\theta) \approx \sum_{i=1}^{C} p_i \cdot [JJ^\top]_{ii} - p^\top (JJ^\top) p$$

$$\mathrm{Tr}(H_\theta) \approx \sum_{i=1}^{C} p_i \sum_{l=1}^{L} \|a^{(l-1)}\|_2^2 \cdot \delta_i^{(L)\top} M^{(l)} \delta_i^{(L)} - \sum_{i,j=1}^{C} p_i p_j \sum_{l=1}^{L} \|a^{(l-1)}\|_2^2 \cdot \delta_i^{(L)\top} M^{(l)} \delta_j^{(L)}$$

Let $s_{l-1} = \|a^{(l-1)}\|_2^2$, and denote $v_i = \delta_i^{(L)}$. Then:

$$\boxed{\mathrm{Tr}(H_\theta) \approx \sum_{l=1}^{L} s_{l-1} \left( \sum_{i=1}^{C} p_i \cdot v_i^\top M^{(l)} v_i - \sum_{i,j=1}^{C} p_i p_j \cdot v_i^\top M^{(l)} v_j \right)}$$

Since $M^{(l)}$ is positive semi-definite and $s_l \geq 0$, we can bound each layer's contribution using the spectral norm:

$$v_i^\top M^{(l)} v_i \leq \lambda_{\max}\left(M^{(l)}\right) \cdot \|v_i\|^2 (Rayleigh - quotient)$$

$$v_i^\top M^{(l)} v_j \geq -\lambda_{\max}\left(M^{(l)}\right) \cdot \|v_i\| \cdot \|v_j\| (Cauchy - Schwarz)$$

Thus, a simple upper bound is:

$$\mathrm{Tr}(H_\theta) \leq \sum_{l=1}^{L} s_{l-1} \cdot \lambda_{\max}\left(M^{(l)}\right) \cdot \left( \sum_{i=1}^{C} p_i \|v_i\|^2 + \sum_{i,j=1}^{C} p_i p_j \|v_i\| \|v_j\| \right)$$

Recognizing that $\sum_{i,j} p_i p_j \|v_i\| \|v_j\| = \left(\sum_i p_i \|v_i\|\right)^2 \leq \sum_i p_i \|v_i\|^2$ (by Jensen's inequality), we obtain the final, clean upper bound:

$$\boxed{\mathrm{Tr}(H_\theta) \leq 2 \sum_{l=1}^{L} \|a^{(l-1)}\|_2^2 \cdot \lambda_{\max}\left(M^{(l)}\right) \cdot \sum_{i=1}^{C} p_i \|\delta_i^{(L)}\|^2}$$

## A.2 Impact of the Softmax Distribution on the Hessian: Derivation of the Variance Identity

We aim to prove the following identity:

$$\sum_{i=1}^{C} p_i \|v_i - \bar{v}\|^2 = 1 - \|p\|_2^2$$

where $v_i = e_i$ denotes the standard basis vector in $\mathbb{R}^C$ (a consequence of backpropagation through the final layer), and $\bar{v} = \sum_{j=1}^{C} p_j v_j = \sum_{j=1}^{C} p_j e_j = p$ is the expectation over the predicted probability distribution $p = (p_1, \ldots, p_C)$.

For each class $i$, the squared Euclidean distance is given by

$$\|e_i - p\|^2 = \sum_{k=1}^{C} ((e_i)_k - p_k)^2$$

Since $(e_i)_k = 1$ if $k = i$ and 0 otherwise, we have

$$\|e_i - p\|^2 = (1 - p_i)^2 + \sum_{k \neq i}(0 - p_k)^2 = (1 - p_i)^2 + \sum_{j \neq i} p_j^2$$

We now compute the weighted sum:

$$\sum_{i=1}^{C} p_i \|e_i - p\|^2 = \sum_{i=1}^{C} p_i \left( (1 - p_i)^2 + \sum_{j \neq i} p_j^2 \right)$$

This expression can be decomposed into two terms:

$$= \underbrace{\sum_{i=1}^{C} p_i(1 - p_i)^2}_{\text{Term A}} + \underbrace{\sum_{i=1}^{C} p_i \sum_{j \neq i} p_j^2}_{\text{Term B}}$$

Term A: $\sum_i p_i(1 - p_i)^2$

Expanding the square yields

$$(1 - p_i)^2 = 1 - 2p_i + p_i^2$$
$$p_i(1 - p_i)^2 = p_i - 2p_i^2 + p_i^3$$
$$\text{Term A} = \sum_{i=1}^{C}(p_i - 2p_i^2 + p_i^3) = \sum_{i=1}^{C} p_i - 2\sum_{i=1}^{C} p_i^2 + \sum_{i=1}^{C} p_i^3 = 1 - 2\|p\|_2^2 + \|p\|_3^3,$$

where we used $\sum_i p_i = 1$, $\|p\|_2^2 = \sum_i p_i^2$, and $\|p\|_3^3 = \sum_i p_i^3$.

Term B: $\sum_i p_i \sum_{j \neq i} p_j^2$

By interchanging the order of summation,

$$\sum_{i=1}^{C} p_i \sum_{j \neq i} p_j^2 = \sum_{j=1}^{C} p_j^2 \sum_{i \neq j} p_i$$

For fixed $j$, the inner sum satisfies $\sum_{i \neq j} p_i = 1 - p_j$. Thus,

$$\text{Term B} = \sum_{j=1}^{C} p_j^2(1 - p_j) = \sum_{j=1}^{C} p_j^2 - \sum_{j=1}^{C} p_j^3 = \|p\|_2^2 - \|p\|_3^3$$

Summing both terms,

$$\sum_{i=1}^{C} p_i \|e_i - p\|^2 = \text{Term A} + \text{Term B}$$

$$= \left(1 - 2\|p\|_2^2 + \|p\|_3^3\right) + \left(\|p\|_2^2 - \|p\|_3^3\right)$$

$$= 1 - \|p\|_2^2$$

**Final Result**

$$\boxed{\sum_{i=1}^{C} p_i \|v_i - \bar{v}\|^2 = 1 - \sum_{i=1}^{C} p_i^2 = 1 - \|p\|_2^2}$$

This identity reveals a geometric interpretation of prediction uncertainty in the output space: the weighted average of squared distances from one-hot vectors $e_i$ to their mean $p$ equals $1 - \|p\|_2^2$, which is precisely the *Gini impurity* of the distribution $p$:

$$\text{Gini}(p) = \sum_{i=1}^{C} p_i(1 - p_i) = 1 - \sum_{i=1}^{C} p_i^2$$

Thus, the quantity $\sum_i p_i \|v_i - \bar{v}\|^2$ serves as a geometric measure of *inter-class gradient dispersion* or *decision uncertainty* in the softmax output space. It attains its maximum under uniform prediction ($p_i = 1/C$) and its minimum under deterministic prediction ($p$ is one-hot), aligning with intuitive notions of model confidence and loss landscape flatness.

## A.3 Synergistic Changes in Sharpness Activation Frequency and Gradient Norm

We will now show the synergistic changes observed during the training process. For MNIST, when using an FFN, the activation frequency, the gradient norm of the last layer, and the Trace of the Hessian all show nearly identical trends. The co-occurrence of activation frequency and Hessian was only observed in the MNIST and FFN scenario. We are unsure of the exact reason for this Figure 7. We use the results of RoBERTa as a control. Figure 8

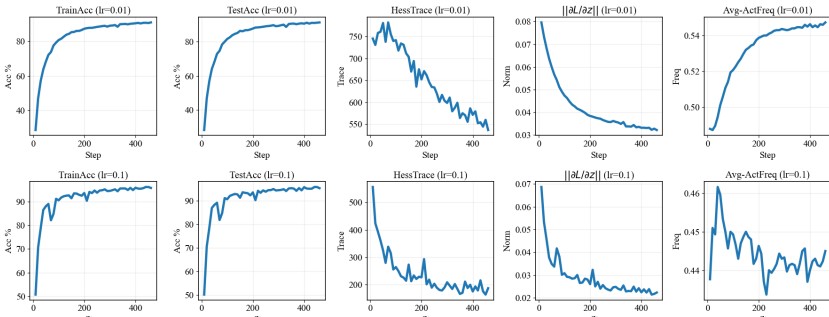

Figure 7: Trained for 1 epoch on MNIST using FFNs with a learning rate of 0.01 (first line) and 0.1 (second line), and momentum = 0. The network used the ReLU activation function with a 128-dimensional hidden layer. From left to right: training set accuracy and test set accuracy, Trace of the Hessian, norm of the initial gradient of backpropagation and the average activation frequency across all layers.

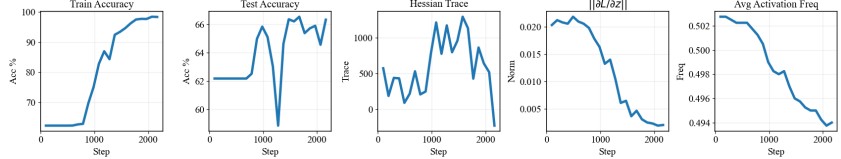

Figure 8: Trained for 7 epoch on boolq using RoBERTa with a learning rate of 2e-5 on Adam.We use a downsized version of RoBERTa with 21M parameters and ReLU activation functions, and conduct experiments on the BoolQ dataset. In the figure, from left to right: training accuracy, test accuracy, Hessian trace, gradient norm of the last layer, and activation frequency (averaged across all layers).

## A.4 Experiment on Feature Map Death in ResNet-18

Table 5: Performance comparison between Mask and UnMask versions on Resnet- 18. Using SGD(lr = 0.01 , momentum = 0.9).

| Version | Train Acc (%) | Test Acc (%) | Tr($\mathbf{H}$) | $\mathbf{H}\lambda_{\max}$ |
|---|---|---|---|---|
| Mask 70% | 89.86 | 65.50 | **2863.1796** | **155.6508** |
| | 90.95 | 65.42 | **2878.2399** | **149.2074** |
| UnMask | 89.98 | 69.53 | **3421.8131** ↑ | **132.2791** |
| | 92.02 | 69.78 | **3508.5457** ↑ | **131.1579** |

We observe that, on ResNet-18, the masked version still has a smaller Hessian trace and worse generalization performance compared to the non-masked version. However, the reason why the largest eigenvalue shows no significant difference remains unclear, the largest eigenvalue only reflects sharpness along a single direction and does not represent the overall sharpness. The underlying mechanism warrants further investigation.

## A.5 TRAINING DYNAMICS WITH NOISE INJECTION ON LARGER DATASETS AND QUANTITATIVE MONITORING

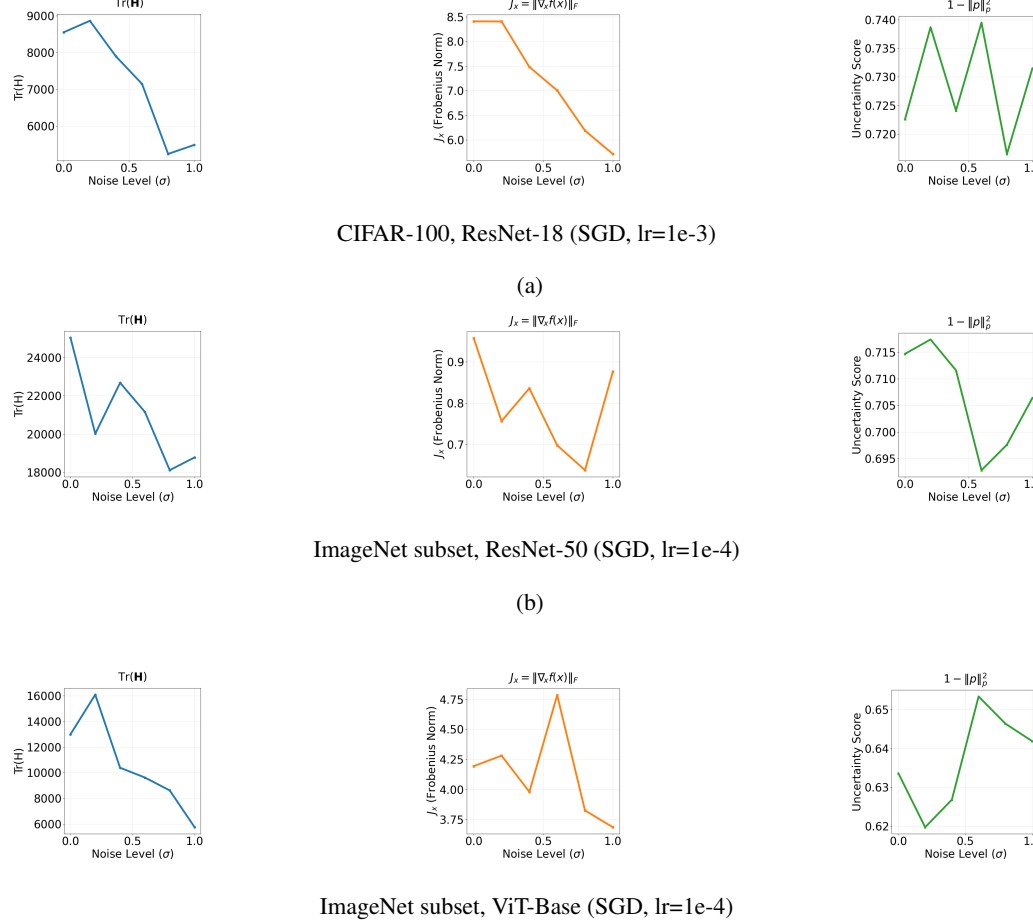

CIFAR-100, ResNet-18 (SGD, lr=1e-3)

(a)

ImageNet subset, ResNet-50 (SGD, lr=1e-4)

(b)

ImageNet subset, ViT-Base (SGD, lr=1e-4)

(c)

Figure 9: After the first epoch, monitoring the effect of Gaussian noise with varying standard deviations ($\sigma$) on model geometry. **Each row** shows results for a different model-dataset pair. **Columns:** Trace of Hessian (left), Input-output Jacobian Frobenius norm (middle), $1-\|p\|_2^2$ (right). All models apply strict per-sample mean-variance normalization as described in the main text.

Here, we additionally examine the Jacobian norm of the input-output mapping and the quantity $1 - \|p\|_2^2$. The former serves merely as an exploratory or curiosity-driven experiment, as noise injection may also affect the sensitivity of the input-output relationship. The latter is introduced to demonstrate that noise injection is unrelated to the softmax output distribution discussed earlier.

We observe that for the CIFAR-100 and ResNet-18 group, the results are highly consistent: as noise injection reduces the Jacobian norm, the trace of the Hessian $\mathrm{Tr}(H)$ also decreases, while $1-\|p\|_2^2$ fluctuates, suggesting that it can be approximately treated as unaffected. However, on the ImageNet subset with ResNet-50, although $\mathrm{Tr}(H)$ still decreases, both $1 - \|p\|_2^2$ and the Jacobian change simultaneously.

In the case of ImageNet with Vision Transformer (ViT), while $\mathrm{Tr}(H)$ decreases, the correlation between $\mathrm{Tr}(H)$, $1 - \|p\|_2^2$, and the Jacobian becomes less clear.

## A.6 AN ALTERNATIVE EXPLANATION FOR FLATTER LOSS LANDSCAPES UNDER NOISE INJECTION

A previous work, known as Sharpness-Aware Minimization (SAM) Foret et al. (2021), introduces a training paradigm that goes beyond standard empirical risk minimization. The method can be summarized as follows:

$$\min_{\theta} \mathcal{L}(\theta)$$

SAM minimizes the maximum loss within a neighborhood of the parameters:

$$\min_{\theta} \max_{\|\epsilon\|_2 \leq \rho} \mathcal{L}(\theta + \epsilon)$$

Due to the high computational cost of solving this min-max problem exactly, SAM employs an efficient approximation:

**Step 1: Find the worst-case perturbation $\epsilon$**

At current parameters $\theta$, find the small perturbation within the ball of radius $\rho$ that maximally increases the loss:

$$\hat{\epsilon} = \arg\max_{\|\epsilon\|_2 \leq \rho} \mathcal{L}(\theta + \epsilon) \approx \rho \cdot \frac{\nabla_{\theta}\mathcal{L}(\theta)}{\|\nabla_{\theta}\mathcal{L}(\theta)\|_2}.$$

**Step 2: Update parameters using the gradient at the perturbed point**

Compute the gradient at the perturbed location and use it for updating:

$$g_{\text{SAM}} = \nabla_{\theta}\mathcal{L}(\theta + \hat{\epsilon}),$$
$$\theta \leftarrow \theta - \eta g_{\text{SAM}}.$$

Re-examining this procedure, we note that instead of explicitly seeking the direction of maximal sharpness, an isotropic exploration within the $\epsilon$-ball would achieve a similar effect since the region of intersection between the $\rho$-radius ball and the loss landscape must include components along the sharpest directions. Under such a reformulation, SAM effectively becomes a form of *noise injection during optimization*. The update can then be rewritten as:

$$\theta \leftarrow \theta - \eta \mathbb{E}_{\epsilon \sim B_{\rho}(0)}\left[\nabla_{\theta}\mathcal{L}(\theta + \epsilon)\right]$$

or approximately,

$$\theta \leftarrow \theta - \frac{\eta}{K}\sum_{k=1}^{K}\nabla_{\theta}\mathcal{L}(\theta + \epsilon_k), \quad \|\epsilon_k\|_2 \leq \rho$$

While it remains unclear whether SAM's success stems from this implicit noise injection, the question of whether sharpness and generalization are causally or merely correlatively linked is still open. Notably, similar effects can be achieved directly in the input space. Consider an input image as a vector $x \in \mathbb{R}^d$, and add Gaussian noise $s \sim \mathcal{N}(\mu, \sigma^2 I)$, resulting in $x + s$. With sufficient sampling, this constructs, in input space, a **radius-$\sigma$, anisotropic ellipsoid**, which we denote as $p$. Through the multi-layer nonlinear transformation of a neural network, $p$ is mapped to a distorted topological structure $p'$, and the final output becomes $y + p'$. Differentiating $\mathcal{L}(y + p')$ with respect to the input noise reveals that the effect propagates to the output as a warped version of the original ball. Such a process can also achieve the effect of SAM, leading to a flatter loss landscape.

This naturally leads to a crucial question: is the relationship between sharpness and generalization causal or merely correlative? Can noise injection achieve the same effect as SAM? And could the improved generalization arise from noise injection enhancing the smoothness of the inference pathway? Yuan et al. (2023) These questions remain open and warrant further investigation.

## B APPENDIX

### B.1 MORE RELATED WORK

Since we have already covered a substantial body of prior work in the main text, and because our conclusions differ significantly from much of the existing literature, we have placed an expanded "More Related Work" section in the appendix due to space constraints.

We demonstrated in the Introduction how scaling network weights affects the Hessian. However, a natural question arises: is weight scale truly unrelated to generalization? One might reasonably ask: if weight decay improves generalization, why would we arbitrarily rescale weight magnitudes? Fortunately, prior work has already addressed this: weight decay does not alter the scale of weights per se, but rather modifies the optimization trajectory acting as a regularization mechanism on the optimization process, equivalent to an anisotropic learning rate scheduling strategy. This is precisely the conclusion established in D'Angelo et al. (2024).

In fact, there are many theoretical measures of generalization, such as VC dimension, PAC generalization bounds, and Rademacher complexity. Among these, PAC-Bayes is directly connects to the notion of sharpness we discussed. These theories are indeed elegant from a mathematical standpoint, but in practice, they often feel frustrating when applied to deep learning: they tend to be naively borrowed from classical learning theory without proper adaptation. As demonstrated in Jiang et al. (2019), large-scale empirical studies have shown that many of these complexity measures can be unreliable or even misleading in modern deep networks. Subsequent work has explored how to design better, more predictive generalization metrics Dziugaite et al. (2021).

For our Hessian-based sharpness measure, in addition to the scaling factors discussed in the main text, a subsequent study Kwon et al. (2021) proposed a scale-invariant sharpness metric and introduced an improved variant of Sharpness-Aware Minimization (SAM) called ASAM. While this work addresses parameter-scale sensitivity, it does not explain the other counterexamples we present. More importantly, that approach focuses solely on parameter scaling, whereas our theoretical analysis and empirical results demonstrate that all scales matter particularly the implicit changes in forward activation norms induced by variations in activation frequency. Although Hessian-based sharpness intuitively reflects sensitivity to parameter perturbations, the ultimate dependent variable is the network's output, which is inevitably influenced by the magnitude of activations throughout the forward pass. Therefore, any sharpness measure that ignores these norm dynamics provides an incomplete picture of generalization behavior.

However, extending our findings to other domains, remains an open question. For instance, Milling et al. (2024) explores a modified sharpness-aware approach in the context of audio. Whether our conclusions continue to hold in such settings is yet to be determined and warrants further investigation.

## B.2 VGG-SMALL MODEL ARCHITECTURE

Table 6: Custom VGG-small

| Layer | Operation | Output Shape |
|---|---|---|
| **Input** | | $(3, H, W)$ |
| **Layer 0** | Conv2d(3 → 64, kernel=3, pad=1) | $(64, H, W)$ |
| **Layer 1** | RecordReLU | $(64, H, W)$ |
| **Layer 2** | MaxPool2d(stride=2) | $(64, H/2, W/2)$ |
| **Layer 3** | Conv2d(64 → 128, kernel=3, pad=1) | $(128, H/2, W/2)$ |
| **Layer 4** | RecordReLU | $(128, H/2, W/2)$ |
| **Layer 5** | MaxPool2d(stride=2) | $(128, H/4, W/4)$ |
| **Layer 6** | Conv2d(128 → 256, kernel=3, pad=1) | $(256, H/4, W/4)$ |
| **Layer 7** | RecordReLU | $(256, H/4, W/4)$ |
| **Layer 8** | MaxPool2d(stride=2) | $(256, H/8, W/8)$ |
| **Layer 9** | Conv2d(256 → 512, kernel=3, pad=1) | $(512, H/8, W/8)$ |
| **Layer 10** | RecordReLU | $(512, H/8, W/8)$ |
| **Layer 11** | MaxPool2d(stride=2) | $(512, H/16, W/16)$ |
| **Global Pooling** | AdaptiveAvgPool2d(2×2) | $(512, 2, 2)$ |
| **Flatten** | | 2048 |
| **Classifier** | Linear(2048 → num_classes, bias=False) | (num_classes,) |

