# OpenReview forum: "Sharpness Can Be Manipulated and Misleading for Generalization"
_ICLR.cc/2026/Conference — ICLR 2026 Conference Withdrawn Submission_

### Official Review · Reviewer_z8sy · 2025-10-29

**Soundness:** 2
**Presentation:** 2
**Contribution:** 1
**Rating:** 2
**Confidence:** 4

**Summary:**

The authors investigate the relationship between sharpness and generalization. In particular, they provide two expressions for the trace of the Hessian, an approximation by focussing on the Gauss-Newton part of the Hessian and neglecting the residual part, and an upper bound of this approximation based on the spectral norm and Jensens inequality. From their expressions, they identify factors that might impact sharpness: weight and activation norms, activation frequency, the entropy of the softmax distribution. They provide experiments with interventions targeting those factors, where sometimes higher sharpness leads to better-performing models (and vice versa). The authors conclude that sharpness measures can be misleading, as they can be manipulated, potentially without affecting the generalization performance.

**Strengths:**

- the experiments are creative
- the upper bound is novel (to the best of my knowledge)
- that finetuning with noise can alter the trace of the hessian without affecting generalization is an interesting observation

**Weaknesses:**

1. Writing and Clarity:
I find the paper hard to read and to follow. I see several reasons for this: i) The central ideas are communicated poorly, ii) often, it is unclear which message a certain Section should convey, and how Sections are connected, iii) the captions of the Figures typically do not explain the message of the Figures, iv) Section 3 introduces the expressions without any context, and it is unclear to the reader where the upper bound and the centered decomposition come from.


2. Doubts about expressions:
The authors already state that the upper bound is “relatively loose”, yet claim, that it “remains informative”. It is unclear why it would remain informative, if the bound is loose. Further, there is work claiming that the residual part of the Hessian should not be neglected when analyzing sharpness [1]. Furthermore, for their analysis in Section 5, the authors set the backpropagation matrix M to the identity matrix. I do not see reason why this would be a valid approximation: In my understanding, M captures all the weight and activation derivatives between some layer and the output logits, and by setting it to the identity matrix, one is effectively assuming the network does not exist.


2. Inconclusive experiments and logical fallacies:
I find many of the experiments and arguments inconclusive. For instance,
- in section 3.2 the authors argue about how weight norms and activation norms and sparsity might affect the bound, but also admit that they might indirectly influence the $\lambda_{max}$, so the overall effect on the bound is unclear
-  Appendix A3 shows that the results about activation frequency are inconclusive, as there, the Hessian trace decreases with increasing activation frequency in once case (Fig. 7), and shows no clear trend with decreasing activation frequency in another case (Fig. 8)
- similarly, Appendix A4 shows that the mask / unmask experiments show no clear trend for the ResNet-18: while the hessian trace decreases for the masked case, lambda-max increases, again not revealing a clear trend
- Section 5 aims to argue that, given controlled network norms, the sharpness can be manipulated by changing the entropy of the softmax / impurity. However, I struggle to find evidence for this from the results in Section 5.2, as i) for the first three rows in Table 2, sharpness changes strongly, but the impurity barely changes, ii) data augmentation leads to better generalization and lower sharpness (as expected by the conventional sharpness-generalization assumption), iii) by increasing R from 1 to 5 (90 epoch setting), generalization improves and sharpness decreases. Overall, I see no evidence for a causal relation between the impurity and sharpness from those experiments
- Section 5.3 investigates overfitting via data leakage  in order to create overconfident outputs. However, the causal chain in the arguments seems flawed to me: By introducing the data leakage, the model is essentially solving a different task (MNIST instead of CIFAR classification). Why should we think that we can draw any conclusions from this, other than that an MNIST model might be flatter than a CIFAR model? It is also not surprising that the confidence for MNIST classification is higher, as it is a simpler task.


3. Experimental setup:
The presented experiments are typically done with a single model and dataset, and a single seed, and thus limited in the conclusions that can be drawn from them. For the cases where different setups are investigated (e.g. Fig 7 and 8), there is often no clear trend, questioning the reproducibility of the presented results in other setups. Overall, the investigated setup (mostly VGG-small on very easy datasets, without BatchNorm) is extremely simple, and it is unclear how the results would translate to more realistic settings.

4. Novelty:
I see the novelty of this work as very limited. That Sharpness can be manipulated has been shown before [3], that weight scale plays a role has been shown [3,4,5], and that logit normalization might be relevant has also been discussed [4,6]. Finally, there is extensive experimental evidence of examples where sharpness does not correlate well with generalization [6].


Minor remarks:
- Table 2: Why mention N, if N=0 for all datapoints, and what are the rows? The first three rows are after epoch one, for the others it is unclear (after epoch 90?). Also, why is DA only specified for two rows?
- it would be good to report test accuracy in Table 3
- often there is either the trace or the lambda-max reported, and it is unclear to me why one or the other and not both are reported
- in Figure 5, it is not clearly indicated what is with and without normalization
- in 6.2: the authors claim that 99% train accuracy is achieved, yet 89.22 is reported in the Table
- “According to our theory, it is reasonable to hypothesize that in the early stages of training, the model tends to memorize the data, leading to a strong dependence between inputs and outputs and consequently a sharper loss landscape” it is unclear to me how this is derived from the provided theory
- Appendix A6: “Can noise injection achieve the same effect as SAM?” This has been studied, e.g. in the original SAM paper, or in [2]. While random perturbations can bring some improvements, they are typically significantly smaller than gradient or momentum-based perturbations.



[1] Dauphin, Y., Agarwala, A., and Mobahi, H. Neglected hessian component explains mysteries in sharpness regularization. In The Thirty-eighth Annual Conference on Neural Information Processing Systems, 2024

[2] Becker et al, Momentum-SAM: Sharpness Aware Minimization without Computational Overhead, https://arxiv.org/pdf/2401.12033

[3] Laurent Dinh, Razvan Pascanu, Samy Bengio, and Yoshua Bengio. Sharp minima can generalize for deep nets, 2017

[4] Tsuzuku, Y., Sato, I., and Sugiyama, M. Normalized flat minima: Exploring scale invariant definition of flat minima for neural networks using pac-bayesian analysis.

[5] Kwon et al, ASAM: Adaptive Sharpness-Aware Minimization for Scale-Invariant Learning of Deep Neural Networks, ICML 2021

[6]  Maksym Andriushchenko, Francesco Croce, Maximilian Müller, Matthias Hein, and Nicolas Flammarion. A modern look at the relationship between sharpness and generalization.

**Questions:**

I struggle to see the contributions of this paper beyond existing work. While the authors do present experimental results where models with sharper minima generalize better (e.g. via noise finetuning or masking activations), similar results have been shown in the past. In other words, that there are cases where sharper minima generalize better than flatter minima is extensively documented. Further, the connection to their expressions regarding the trace of the hessian, and causal links to the derived components are questionable, as experimental results show correlation at best, and even there the results are often mixed. Finally, the experimental setup is often extremely simple, and the paper is hard to read and process.

---

### Official Review · Reviewer_XxGB · 2025-10-30

**Soundness:** 1
**Presentation:** 1
**Contribution:** 2
**Rating:** 2
**Confidence:** 4

**Summary:**

The authors attempt to disentangle the commonly assumed causal relationship between flatness and generalisation; they use a series of experimental setups that explore the impact of activation frequency, feature map masking, softmax distribution, repeated training batches, data leakage and noise injection on Hessian-based sharpness measures. For their empirical experiments, they consider a range of different architectures, including VGG-small, ResNet-18, and a ViT primarily employing the MNIST and CIFAR (10 and 100) and a subset of the ImageNet dataset. Through their theoretical and empirical analysis, their core finding is that Hessian-based measures can be manipulated, suggesting that flatness may be correlated with generalisation, but not a causal factor for it.

**Strengths:**

1. The motivation of the study is interesting, and exploring different perspectives on existing theories around sharpness helps to provide new insights into deep learning and generalisation.
2. The study considers a range of different empirical experiments (approximately 6) to provide counterexamples for the causal relation suggested with generalisation and flatness (Hessian-based) measures, some of which are quite interesting.
3. Showing the impacts of different experimental conditions across different datasets and architectures is an important aspect of the work.
4. The study of activation frequency and generalisation is interesting, and I have not seen this angle explored in current literature; however, I feel the results in this section could be strengthened (please see weaknesses below).

**Weaknesses:**

1. **Sharpness measures**: The central question of the paper 'Is Hessian-based sharpness truly a reliable indicator of generalization?' does not appear to me to be a novel one. Indeed, the work done by Petzka et al. [1] shows that Hessian-based flatness measures using the trace of the loss-Hessian are not reparameterisation invariant and, therefore, not a reliable indicator of generalisation. [1] does, however, show that Relative Flatness [1] and Fisher Roa norm [2] are Hessian-based measures which are reparameterization invariant. As a result, the focus on measures such as the Trace of the Hessian ($Tr(H)$) and the Max Hessian ($Hλ_{max}$) appears odd, as for at least one of these metrics, it is known that it is not a reliable indicator of generalisation. As a result, I would like to see experiments conducted looking at the experimental setups' impact on the Relative Flatness and Fisher Roa norm measures of sharpness to confirm the findings observed in the paper.

2. **No average of results or Standard Error of the Mean reported**: In all tables and figures presented, there is no average or Standard Error of the Mean [3] reported; this is basic scientific practice and provides an improved reliability of the findings. Without this, it is hard to assess how reliable your empirical experiments are.

3. **Activation Frequency Results**: In section 4.1 from Figure 2, it can be observed that activation frequency can change the value of the Hessian trace; however, I am not so certain about the correlation you identify between the activation frequency and the Hessian Trace. For example, when the perturbation is (approximately) -0.1, the activation frequency increases from the previous point, but it can be observed that the Hessian trace is reduced, showing that an increase in activation frequency does not perfectly describe the increase in Hessian trace. Can you provide a plot showing the impact of perturbation between 1 and 1 in 0.1 intervals? The trends observed are quite sparse with lots of deviation between points, so this would help to better support your points between this relation and make the figure clearer.

4. **Feature Map Masking at 70%**: What motivated the feature Map masking value of 70%? This value feels arbitrary. It would be useful to provide a range of mask percentages, perhaps between 10% to 100% in 10% intervals, that show that as the percentage of masking is increased, $Hλ_{max}$ decreases. Also in Appendix A.4, you present the results for the ResNet18 with the $Tr(H)$ and $Hλ_{max}$  but do not present the $Tr(H)$ in the main body for the masking - is there a reason for this?

5. **Unfair comparison for Figure 3**: Not keeping the training budget equal for your comparison of data augmentation and non-data augmentation does not make sense. Instead, you should provide both models with the same training budget and then compare their sharpnesses at the end of this, such that it is a fair comparison.

6. **Table 2**: The results in Table 2 are hard to parse. What is the rationale behind comparing the sharpness after a single epoch? I see no benefit in doing this, and it takes away from the results below.

7. **Flatness from Noise Injection**: This experiment is very unclear. I believe that a figure would be useful to display what the training samples look like once they have been transformed for this case. Also, I would avoid using terms such as (L402) 'diverging significantly' if no significance tests have been conducted. For Figure 6b, the authors state that (L437) 'when the noise variance exceeds 0.3, training becomes impossible yet sharpness continues to decrease.' but provide no evidence for the lack of model convergence at a noise level of 0.3.

8. **Misc** All figures could be made larger to enable improved readability. Figure 4 has inconsistent labelling. Also, should the label be 'R', as it is in the table, instead of 'inner loop'? Finally, (L439) 's experimental results are provided in Appendix A.4. These experimental results do not seem related to the noise injection results but the masking results - please check this reference is correct. It would be good to include a reference to other works that have also challenged the notion of flatness, such as [4] and [5].

Overall, I find some of the experiments and their results interesting; however, due to the lack of empirical rigour of this work, I find that it requires significant improvements to have an adequate impact on the community. Additionally, focusing on fewer experiments in the main body could allow for improved clarity to fully explain the results/rationale for experiments.

**Questions:**

**Q1**: You state (L215) 'the natural rate of ReLU death in standard convolutional layers is relatively low.' Can you please elaborate on this and provide an explanation as to why you believe that this occurs in convolutional layers and what you think the difference is with other layers, such as MLP layers?

**Q2**: In Appendix A.4, you state that 'largest eigenvalue only reflects sharpness along a single direction and does not represent the overall sharpness.' Why then do you consider this metric throughout the paper if it does not truly represent the overall sharpness of the model? Would it not be better to use other metrics that do?

**Q3**: What is the motivation behind using the ImageNet subset over TinyImageNet [6]?

References:

[1] Petzka, H., Kamp, M., Adilova, L., Sminchisescu, C. and Boley, M., 2021. Relative flatness and generalization. Advances in neural information processing systems, 34, pp.18420-18432.

[2 Liang, T., Poggio, T., Rakhlin, A. and Stokes, J., 2019, April. Fisher-rao metric, geometry, and complexity of neural networks. In The 22nd international conference on artificial intelligence and statistics (pp. 888-896). PMLR.]

[3] Belia, S., Fidler, F., Williams, J. and Cumming, G., 2005. Researchers misunderstand confidence intervals and standard error bars. Psychological methods, 10(4), p.389.

[4] Wen, K., Li, Z. and Ma, T., 2023. Sharpness minimization algorithms do not only minimize sharpness to achieve better generalization. Advances in Neural Information Processing Systems, 36, pp.1024-1035.

[5] Mason-Williams, I., Ekholm, F. and Huszár, F., 2024. Explicit regularisation, sharpness and calibration. In NeurIPS 2024 Workshop on Scientific Methods for Understanding Deep Learning.

[6] Le, Y. and Yang, X., 2015. Tiny imagenet visual recognition challenge. CS 231N, 7(7), p.3.

---

### Official Review · Reviewer_74Lk · 2025-10-30

**Soundness:** 2
**Presentation:** 2
**Contribution:** 2
**Rating:** 4
**Confidence:** 4

**Summary:**

This paper investigates the rumored link between sharpness (as measured by Hessian trace or maximum eigenvalue) and generalization, and argues that this link may be correlational rather than causal.  First, in section 3, the authors present both an approximate upper bound and an approximate equality for the Hessian trace, which may yield insight into the factors that make Hessian trace large or small.  Then, in section 4.1, they show that taking a trained neural net and making the first-layer biases larger makes the Hessian trace smaller while making the test accuracy worse.  Then in section 4.2, they show that artificially masking out some units during training results in smaller Hessian trace as well as worst test accuracy.  Then in section 5.1, they show that data augmentation makes the Hessian trace smaller while making generalization better (I didn't understand how this relates to their core argument).  I don't understand section 5.2.  Then in section 5.3, they show that they can manipulate the input data distribution in such a way as to make the Hessian trace of the learned network smaller while generalization is worse.  Finally in section 6, they show that by injecting noise into the inputs they can make Hessian trace smaller without improving generalization.

**Strengths:**

I appreciated the decomposition of the Hessian trace in section 3.3; I could see it being useful for subsequent owrk.

**Weaknesses:**

* The submission felt quite disorganized.  It was hard to follow the logical flow much of the time.  It was also often hard to understand what experiment was being depicted.

  * I don't understand how the section 4.1 experiment casts doubt on sharpness, given that the train accuracy also decreases as we increase the bias.

  * I don't understand what is being argued in section 5.1.

  * I don't understand what experiment is being run in section 5.2.

* It has already been widely believed that the link between sharpness and generalization is, in many cases, correlational rather than causal.  For example, much like the current submission, the following work identifies settings where some training intervention results in lower sharpness (as quantified by maximum eigenvalue) but not in improved generalization.
> Simran Kaur, Jeremy Cohen, Zachary C. Lipton. "On the Maximum Hessian Eigenvalue and Generalization." ArXiv 2023.

 So I'm not sure if the current submission adds a lot that is new to the conversation.

**Questions:**

Minor:
 - the upper bound in section 3.2 is not a real upper bound, as it relies on some approximations -- you should be clearer about this
 - for the two equations at the bottom of page 3, one does a sum over $s_\ell$ and the other does a sum over $s_{\ell-1}$ -- it seems that at least one of them is a typo?
 - in section 6.2, you write that you trained the ResNet till 99% accuracy, but the table shows the standard train accuracy at 89%.  What's the reason for this discrepancy?

The following paper also has a discussion of the relationship between curvature and the predictive probability distribution, including a discussion of Gini impurity:

  Sungyoon Lee, Jinseong Park, and Jaewook Lee.  "Implicit Jacobian Regularization Weighted with Impurity of Probability Output."  ICML '23.

---

### Official Review · Reviewer_vAjj · 2025-11-01

**Soundness:** 3
**Presentation:** 3
**Contribution:** 3
**Rating:** 6
**Confidence:** 2

**Summary:**

This paper challenges the widely-held relationship between flat minima and generalization. The authors construct counterexamples based on two analytical expressions for the trace of the Hessian: 1) Scale-AWare Upper Bound 2) Centered Decomposition. (1) is governed by the L2 norm of the activations and gradient error (from the output) propagated to each layer and the strength of the backward signal (\lambda_max of M), which directly shows that scale of the weights change sharpness. (2) constructs the Hessian trace as a weighted variance of class-specific gradient signals, where the weights are the softmax probabilities (p_i) coupled with backward dynamics (M).

**Strengths:**

- The paper is clear and well-written, the authors identify interesting counterexamples.
- While I am not particularly versed in the sharpness literature, I believe the proposed upper-bound adds to the existing literature challenging the flatness and generalization claims. The authors systematically evaluate counter-examples motivated by the individual components of this estimate.
- The observation that sharpness is intrinsically linked to the entropy of the softmax distribution—(section 5\) as outputs approach one-hot vectors (high confidence), the landscape becomes flatter. This provides a clear mechanistic explanation for why overconfident models can simultaneously exhibit flat minima and poor generalization.
- The data leakage and noise injection experiments provide insightful counter-examples to existing claims in the field.

**Weaknesses:**

Although I find the hypothesis interesting, I am not quite convinced by the cycling experiment in 5.2. A more compelling experiment could test whether this effect holds for hard batches (e.g., measured by high gradient norm or high loss) or on a dataset with more classes (e.g. Cifar-100), where the model struggles to fit the data. If cycling on difficult batches still produces flat minima without generalization improvements, it would more convincingly demonstrate that flatness can be decoupled from meaningful learning

**Questions:**

- Regarding 4.1: Could the authors include an empirical observation of how often such changes in the activation frequency arises in training?
- What does this imply for model calibration? If flatter minima correspond to one-hot (overconfident) predictions, does this mean models in flatter minima require more calibration than those in sharper minima?
- Could the authors provide empirical validation of how tight the Scale-Aware Upper Bound is in practice?

---

### Note · Authors · 2025-11-12

I have read and agree with the venue's withdrawal policy on behalf of myself and my co-authors.